# Inhibition of ATM Increases the Radiosensitivity of Uveal Melanoma Cells to Photons and Protons

**DOI:** 10.3390/cancers12061388

**Published:** 2020-05-28

**Authors:** Rumana N. Hussain, Sarah E. Coupland, Jakub Khzouz, Helen Kalirai, Jason L. Parsons

**Affiliations:** 1Liverpool Ocular Oncology Research Group, Department of Molecular and Clinical Cancer Medicine, William Henry Duncan Building, University of Liverpool, Liverpool L7 8TX, UK; rhussain@liverpool.ac.uk (R.N.H.); S.E.Coupland@liverpool.ac.uk (S.E.C.); khzouz_jakub@yahoo.com (J.K.); H.Kalirai@liverpool.ac.uk (H.K.); 2St Paul’s Eye Clinic, Liverpool University Hospitals Foundation Trust, Prescot Street, Liverpool L7 8XP, UK; 3Liverpool Clinical Laboratories, Duncan Building, Liverpool University Hospitals NHS Foundation Trust, Liverpool L69 3GA, UK; 4Cancer Research Centre, Department of Molecular and Clinical Cancer Medicine, University of Liverpool, 200 London Road, Liverpool L3 9TA, UK; 5Clatterbridge Cancer Centre NHS Foundation Trust, Clatterbridge Road, Bebington CH63 4JY, UK

**Keywords:** ATM, DNA damage, DNA repair, ionizing radiation, protons, uveal melanoma

## Abstract

Treatment of uveal melanoma (UM) is generally successful, with local primary tumour control being at 90–95%. Localized radiotherapy in the form of plaque brachytherapy or proton beam radiotherapy is the most common treatment modality in the UK. However, the basic mechanisms of radiation response, DNA repair and tissue reactions in UM have not been well documented previously. We have investigated the comparative radiosensitivity of four UM cell lines in response to exogenous radiation sources (both X-rays and protons), and correlated this with DNA repair protein expression and repair efficiency. We observed a broad range of radiosensitivity of different UM cell lines to X-rays and protons, with increased radioresistance correlating with elevated protein expression of ataxia telangiectasia mutated (ATM), a protein kinase involved in the signaling and repair of DNA double strand breaks. The use of an ATM inhibitor in UM cell lines enhanced radiosensitivity following both X-ray and proton irradiation, particularly in cells that contained high levels of ATM protein which are otherwise comparatively radioresistant. In proton-irradiated compared with non-irradiated primary enucleated UM patient samples, there was no significant difference in ATM protein expression. Our study therefore suggests that ATM is a potential target for increasing the radiosensitivity of more resistant UM subgroups.

## 1. Introduction

Uveal melanoma (UM) is the most common primary intraocular malignancy, although it is a rare condition with an annual incidence of 5–8 cases per million [1]. In the UK, although primary enucleation is employed in ~30% of all UM cases, localized radiotherapy (e.g., plaque brachytherapy or proton beam therapy (PBT)) is utilized in most patients, with local tumor control being achieved in 90–95% of cases [2]. Despite radiotherapy being the mainstay of UM treatment, studies investigating the radiobiology and radioresistance of this tumor are limited. Early studies conducted using single UM cell lines (SP6.5 and OM431) suggested that the cells were generally radioresistant to γ-radiation up to doses of 3–4 Gy [3,4,5]. Two studies utilizing four primary (OCM-1, Mel202, Mel270 and 92-1) and four metastatic (OMM1, OMM2.2, OMM2.3 and OMM2.6) UM cell lines appear to suggest a broad spectrum of radiosensitivity to X-ray radiation, with no apparent difference as to whether the cells were derived from the primary or metastatic tumor [6,7]. This was supported by a more recent study performed in a range of UM cell lines (SP6.5, Mel270, μ2, TP17, 92.1 and MKT-BR), which again demonstrated broad radiosensitivity to X-ray radiation, which was neither influenced by cell cycle synchronization (asynchronous or confluent cells in G_0_/G1) nor by hypoxic conditions (1% oxygen) pre- and post-irradiation [8]. This study, however, demonstrated the increased effectiveness of high linear energy transfer (LET) carbon ions, compared to low-LET X-rays, in significantly reducing cell survival. With respect to studies analyzing the response of UM cells to PBT, these are very limited, although it has been demonstrated using Mel270 UM cells that cell survival in response to low-LET protons and X-rays is comparable, but that protons appear to specifically impact cell motility [9,10]. Consequently, an analysis of the cellular DNA damage response in UM cell lines to photons and protons has not been previously reported.

Photon (X-ray) and proton radiotherapy are effective at inducing the formation of DNA damage, such as DNA double strand breaks (DSBs), which contribute to their therapeutic effect in cancer cell killing. Despite this, there is still a significant biological uncertainty with protons, particularly due to increases in LET at and around the Bragg peak, where the majority of the radiation dose is deposited [11]. Cells are, however, equipped with enzymes involved in the cellular DNA damage response (DDR) which orchestrate the repair of the DNA damage. The important protein kinases that co-ordinate the signaling cascade and repair of DNA DSBs are ataxia telangiectasia mutated (ATM), ataxia telangiectasia and Rad3 related (ATR), and the DNA-dependent protein kinase (DNA-Pk), which stimulate repair through either non-homologous end-joining (ATM and DNA-Pk) or homologous recombination [12]. Therefore, targeting ATM, ATR and DNA-Pk has been suggested as a strategy to potentiate the cell killing effects of radiotherapy through the inhibition of DNA DSB repair, particularly in solid tumors (e.g., head and neck cancers). Interestingly, in UM, there is recent evidence demonstrating that the DNA-Pk inhibitor, NU7026, can decrease the survival of UM cell lines (OMM1, OMM2.5, Mel270 and MM28) as a monotherapy [13]. Similarly, the survival of UM cells (SOM 157D and SOM 196B) was also significantly reduced using inhibitors of DNA-Pk (NU7026 and NU7441) alone, and there was limited data presented that SOM 196B displayed increased radiosensitivity in the presence of NU7026 [14]. Interestingly, recent data suggest that loss of ATM and ATR has been observed in UM, and has been associated with metastasis [15,16]. Despite this, the efficiency of UM cells to perform DSB repair in response to photon and proton irradiation, and the discovery of DDR inhibitors that synergize with the radiation in effective cell killing, has not been studied in detail.

In this study, we examined the radiosensitivity of UM cell lines following photon (X-ray) and proton irradiation, and identified that the most radioresistant cell lines contain increased ATM protein levels. These UM cell lines can be radiosensitized using an inhibitor targeting ATM. These results suggest a potential therapeutic strategy employing ATM inhibitors, which could be exploited for more effective treatment of UM by radiotherapy.

## 2. Results

### 2.1. Characterisation of the Radiosensitivity of UM Cells and DNA Repair Protein Profiles

There are limited data available analyzing the radiosensitivity of UM cell lines to both protons and photons, and the relationship to DNA repair protein expression. Therefore, we first analysed the levels of key proteins involved in DNA base damage and single strand break repair (poly(ADP-ribose) polymerase-1, PARP-1; X-ray repair cross-complementing protein 1, XRCC1, polynucleotide kinase phosphatase, PNKP and AP endonuclease 1, APE1) and DNA DSB repair (Ku86, DNA-Pk, ATM, ATR, 53BP1 and RAD51) in extracts from four UM cell lines derived from primary (Mel270 and 92.1) and metastatic (OMM1 and OMM2.5) tumors. Variability was observed particularly in the levels of ATM, XRCC1, PNKP and Pol *β* between the cell lines (Figure 1A; Appendix A). However, it was noticeable that OMM2.5 and Mel270 contained higher levels (~1.7–6.0-fold) of ATM compared to OMM1 and 92.1. There was no apparent correlation between differential protein levels and whether the UM cells were derived from a primary (Mel270 and 92.1) or metastatic (OMM1 and OMM2.5) tumor. 

We subsequently analyzed the comparative radiosensitivity of the cells to photon (X-ray) and proton irradiation by clonogenic survival assays. OMM2.5 and Mel270 were ~3.5–6.2-fold (across the dose response) more radioresistant to X-ray irradiation than OMM1 and 92.1 (Figure 1B,C). The same trend was observed in response to proton irradiation, where OMM2.5 and Mel270 were ~1.8–4.3-fold more radioresistant than OMM1 and 92.1 (Figure 1D,E). It should be noted that comparatively higher doses of protons were required to reduce cell survival versus X-ray radiation, due to cells being positioned at the entrance dose of a pristine (unmodulated) beam. Interestingly though, there appeared to be a greater separation of the differences in cell survival between OMM2.5 and Mel270 in comparison to OMM1 and 92.1, specifically in response to relatively lower doses of X-ray radiation (0.5–2 Gy). Nevertheless, these data are supported by statistical analyses, which highlight that survival of the most radiosensitive UM cell line, 92.1, to photons and protons was statistically different from those which were more radioresistant (OMM2.5 and Mel270), but the radiosensitivity was equal to OMM1 (Table 1). It is notable that the most radioresistant cell lines (OMM2.5 and Mel270) were those that contained higher protein levels of ATM involved in DSB repair (Figure 1A).

### 2.2. DSB Repair Kinetics of UM Cells following Photon and Proton Irradiation

We analyzed the comparative kinetics of DNA DSB repair in UM cell lines in response to photon and proton irradiation using neutral comet assays. We observed in response to photons that, surprisingly, all the four UM cell lines analysed (OMM1, OMM2.5, Mel270 and 92.1) were able to largely repair DSBs within a 4 h period post-irradiation (Figure 2A,C). The cell lines also contained similar levels of DSBs at 1–2 h post-irradiation, even though these display differences in overall radiosensitivity to X-rays (Figure 1B). Whilst the repair efficiencies of the UM cell lines was similar, there were slight differences in the relative levels of DSBs induced immediately post-irradiation with X-rays, particularly in OMM2.5 cells which displayed reduced DSB levels (Figure 2C; compare time 0). However, in response to proton irradiation, the cell lines appeared to show differences in their ability to efficiently perform DSB repair. In fact, the OMM1 cells displayed ~1.4–2.1-fold higher levels of DSBs at 1 and 2 h post-irradiation compared to OMM2.5 and Mel270 cells (Figure 2B,D). Additionally, the 92.1 cells also contained ~1.2–1.8-fold, and ~1.8–2.7-fold higher DSB levels at 1 h and 2–4 h post-irradiation, respectively, compared to OMM2.5 and Mel270 cells (Figure 2B,D). This demonstrates that OMM1 and 92.1 have reduced capacity to perform DSB repair following proton irradiation, which correlates with these cell lines being the most radiosensitive to this radiotherapy modality.

### 2.3. Expresson of ATM Protein in UM

Given the range (~1.7–6.0-fold) of different expression of ATM protein in UM cell lines, and apparent links with modulating cellular radiosensitivity to photons and protons (Figure 1A–D), we analysed the expression of ATM in enucleated eyes from UM patients, which were either proton-irradiated (secondary enucleations; *n* = 32) or non-irradiated (primary enucleations; *n* = 20). Examples of the degree of staining (from negative up to 65% staining) observed in cells from UM tissues are shown (Appendix A). There was no statistically significant difference in UM cell immunoreactivity for ATM between the two cohorts (Appendix A). Additionally, ATM expression was not statistically different in proton-irradiated UM that was secondarily enucleated for tumor progression/recurrence versus those that were enucleated for the development of neovascular glaucoma or retinal detachment, although the former group had a tendency towards stronger ATM expression (Appendix A). ATM staining was also assessed for correlation with other factors associated with metastatic disease, including tumor cell type, ciliary body involvement, tumor size, inflammatory cell infiltration and mitotic count. None of these parameters were significantly correlated with ATM expression levels (Appendix A). Finally, there was also no alteration in ATM expression according to the time period between proton treatment and enucleation (*p* = 0.57).

### 2.4. Targeting ATM can Increase the Radiosensitivity of UM Cells to Photon and Proton Irradiation

Inhibition of ATM using the inhibitor KU-59933 alone had no impact on the survival of UM cell lines by clonogenic assays (Figure 3A). This demonstrated that inhibiting ATM had no significant impact on the survival of UM cells as a monotherapy (*p* = 0.33–0.72; one-sample t-test). We consequently analysed the impact of targeting ATM on the radiosensitivity of UM cell lines in response to both photon (up to 4 Gy) and proton (up to 8 Gy) irradiation. We observed that ATM inhibition resulted in increased radiosensitivity of both the OMM2.5 cells (by ~1.7-fold across the dose response) and, more dramatically, the Mel270 cell lines (by ~16-fold), but not the 92.1 cells, to photon irradiation (Figure 3B–F and Table 2). OMM1 cells, following ATM inhibition, appeared to demonstrate increased radioresistance, although only at a high 4 Gy dose of photons. Statistical analysis of the whole data set confirmed that OMM2.5 and Mel270 cell lines were significantly radiosensitized in the presence of the ATM inhibitor (Table 2). Interestingly, in response to protons, only OMM2.5 was not significantly radiosensitized in the presence of the ATM inhibitor (Figure 4A–E and Table 2). The most dramatic effect of ATM inhibition in combination with protons was observed with OMM1, where a ~4.4-fold increase (from 0–4 Gy; comparative to the doses used for photon irradiation) in radiosensitivity was observed. However, Mel270 also displayed a ~1.8-fold elevation in proton radiosensitivity (from 0–4 Gy). These data would broadly suggest that UM cell lines can be radiosensitized following photons and protons through targeting ATM.

## 3. Discussion

The radiobiology of UM has only been subject to limited investigation. Early studies using single UM cell lines suggested that these are generally radioresistant [3,4,5], but more recently, it has been described that UM cell lines display a broad range of radiosensitivity which is not dependent on whether they were derived from a primary or metastatic tumor [6,7]. These studies have largely been performed using low-LET X-rays/γ-rays. However, one study showed the impact of high-LET carbon ions in significantly reducing the survival of UM cells [8]. Also, more recently, it was described that Mel270 cells display comparative cell survival in response to low-LET protons and X-rays [9,10]. With the mainstay of eye-sparing treatments focusing on Ru^106^ brachytherapy (in Europe) and proton beam therapy for UM, we have utilized both X-rays and protons to investigate the comparative radiobiology of four UM cell lines.

As demonstrated in previous studies, our data support the variability in radiosensitivity between UM cell lines, and that there is no correlation with the UM cell line derivation, be it from a primary or a metastatic tumor. A novel finding of our study is that this degree of variability between UM cell lines is observed following both photon (X-ray) and proton irradiation. In fact, the OMM2.5 and Mel270 cell lines were consistently the most radioresistant, whereas the OMM1 and 92.1 were the most radiosensitive following both X-rays and protons. We also showed that cellular radioresistance appears to correlate largely with the levels of ATM that co-ordinates the signaling and repair of DNA DSBs. Indeed, an ATM inhibitor was able to enhance the effects of photons and protons in reducing clonogenic cell survival in a range of UM cell lines. This suggests a potential therapeutic strategy to enhance the impact of radiotherapy (photons and protons) in UM cell killing. Targeting ATM is currently being investigated to enhance the radiosensitivity of other tumor types, particularly glioblastoma [18,19,20], but also head and neck cancers [21], largely focused on photon irradiation. However, we now describe that ATM inhibitors may be more effective in the treatment of UM in response to both photons and protons. We are aware that more potent and selective inhibitors targeting ATM, particularly AZD1390, are now available, and this will be the focus of our future studies, in combination with both photons and protons in enhancing the radiosensitivity of UM cell models. Interestingly, there are recent studies suggesting that the inhibition of another kinase involved in DSB repair, DNA-Pk, as a monotherapy can reduce the survival of UM cell lines [13,14]. There are also very limited data suggesting that DNA-Pk inhibition can work effectively in combination with X-rays in preventing UM cell survival [14]. Despite this, we didn’t observe any dramatic changes in DNA-Pk protein levels in the UM cell lines that showed variations in radiosensitivity utilized in this study. However, given the relatively high cellular levels of DNA-Pk and Ku86, it is possible that minor fluctuations in these protein levels not detectable by immunoblotting may have more dramatic consequences on radiation-induced cell survival. Nevertheless, taken together, these studies and ours suggest that targeting the DDR should be investigated in more detail in UM cell lines to increase radiotherapy efficacy. Additionally, these studies should be expanded to include appropriate 3D models (e.g., spheroids [22]), that more closely mimic the structure and environment of the original tumor that is being treated.

We analyzed the expression of ATM in UM tissues, but did not observe any difference in immunohistochemical staining between proton-irradiated and treatment-naive UM, nor any correlation of ATM staining with known poor prognostic parameters or clinical outcomes. Given the small sample size, it is possible that our study was not powered sufficiently to identify any statistical difference. The phosphorylation of ATM and the relocation of the protein into the nucleus have been suggested to act as predictive biomarkers of radiotherapy response [23,24,25] and prognosis in specific tumors, such as cervical tumors [26]. Consequently, it would be interesting to examine whether activation of ATM itself, and also of its downstream targets (e.g., γH2AX and CHK2), is dysfunctional in UM patients post-irradiation, and the correlation with radiosensitivity and clinical outcome. Interestingly, in a recent study, greater gene expression of DNA-Pk was observed in metastatic UM, and was significantly associated with an adverse clinical outcome [13]. A second study also confirmed greater DNA-Pk expression in UM, although, similar to our study, no changes in ATM expression were observed [14]. This compares to recent data identifying loss of ATM and ATR in UM with a high risk of metastasis [15,16], although it should be noted that the goal of our study was not to determine whether specific DNA repair proteins are biomarkers of metastatic risk. Given these apparent discrepancies, further analysis of the expression of key proteins involved in DSB repair, including DNA-Pk, ATM and ATR, are required in larger UM cohorts.

Interestingly, our analyses found no differences in the kinetics of DSB repair through neutral comet assay analysis in UM cell lines following X-rays, suggesting that there is no direct correlation with DSB repair efficiency and radiosensitivity. This could indicate that the inherent photon (X-ray) radiosensitivity is largely determined by other factors, such as chromatin structure, cell cycle progression, and cell survival mechanisms including apoptosis, autophagy and senescence. This requires further investigation in UM. However, in response to protons, there was a correlation with UM cells displaying reduced DSB repair capacity post-irradiation (OMM1 and 92.1), and their increased radiosensitivity compared to the other cell lines examined (OMM2,5 and Mel270). This could alternatively suggest a different spectrum of DNA damage induced by proton rather than photon irradiation, or that the DSB repair pathways utilized (e.g., NHEJ or HR) are distinct in their response to the different radiation modalities. Therefore, a comprehensive analysis of the induction and repair of DNA base damage and single strand breaks (e.g., via alkaline comet assays), and DSBs (via γH2AX, 53BP1 and RAD51 foci analysis; plasmid-based assays to measure NHEJ/HR efficiency) following photon and proton irradiation is necessary to further understand this. It was noticeable, though, that lower doses of X-ray radiation (0.5–2 Gy) appeared to have a greater impact on reducing cell survival than protons, when comparing the response between OMM2.5 and Mel270 with OMM1 and 92.1. Given the lack of correlation with DSB repair efficiency, the mechanism through which reduced cell survival (e.g., apoptosis or senescence) is triggered in response to X-rays at these lower doses should be investigated in more detail. It should be noted that our proton experiments utilized low-LET protons at the entrance dose of a pristine beam, and therefore we would not expect LET, and subsequently DNA damage complexity, to play a role in the differential cellular response compared to photons. Nevertheless, this was recently discussed in the context of head and neck cancer cells [11], and requires further investigation in UM cell lines, particularly employing DSB repair-specific assays or end-points (e.g., γH2AX/53BP1 or RAD51 foci analysis). In a clinical context, this may reflect the few clinical cases of UM that display poor response to Ru^106^ brachytherapy but which have subsequently been successfully treated with protons.

## 4. Materials and Methods 

### 4.1. Antibodies

The following antibodies for immunoblotting were used. APE1, Pol β and XRCC1 antibodies were kindly provided by Dr G. Dianov. PNKP antibodies were also kindly provided by Prof M. Weinfeld. ATM (sc-23921), DNA-Pk (sc-9051), Ku-86 (sc-9034), PARP-1 (sc-53643) and RAD51 (sc-8349) antibodies were from Santa Cruz Biotechnology (Heidelberg, Germany); ATR antibodies (ab2905) were from Abcam (Cambridge, UK); 53BP1 (A300-272A) antibodies were from Bethyl Laboratories (Montgomery, USA); and actin (A5441) antibodies were from Sigma-Aldrich (Gillingham, UK). 

### 4.2. Cell lines, Culture Conditions and Radiation Sources 

UM cells (OMM1, OMM2.5, Mel270 and 92.1) were routinely cultured as monolayers in RPMI medium supplemented with 10% fetal bovine serum, 2 mM L-glutamine, 1× penicillin-streptomycin and 1× non-essential amino acids. All cells were cultured under standard conditions in 5% CO_2_ at 37 °C, and were authenticated in our laboratory by STR profiling. Cells were irradiated using a CellRad X-ray irradiator (Faxitron Bioptics, Tucson, USA), or at the entrance dose of a passive scattered horizontal proton beam line at 60 MeV maximal energy (~1 keV/µm), as previously described [27,28].

### 4.3. Whole Cell Extracts and Immunoblotting

Whole cell extracts were prepared as previously described [29]. Briefly, UM cells were harvested by centrifugation (1500 rpm for 5 min at 4 °C), and pellets were resuspended in one packed cell volume (PCV) of buffer containing 10 mM Tris-HCl (pH 7.8), 200 mM KCl, 1 μg/mL of each protease inhibitors (pepstatin, aprotinin, chymostatin and leupeptin), 1 mM Phenylmethylsulfonyl fluoride (PMSF) and 1 mM Dithiothreitol (DTT). Two PCV of buffer containing 10 mM Tris-HCl (pH 7.8), 600 mM KCl, 40% glycerol, 0.1 mM ethylenediaminetetraacetic acid (EDTA), Igepal CA-630, 1 μg/mL of each protease inhibitor (pepstatin, aprotinin, chymostatin and leupeptin; Sigma-Aldrich, Gillingham, UK), 1 mM PMSF and 1 mM DTT was added, and the cell suspension was mixed thoroughly prior to mixing by rotation for 30 min at 4 °C. The cell lysate was then centrifuged at 40,000 rpm for 20 min at 4 °C, the supernatant collected, aliquoted and stored at −80 °C. For immunoblotting, cell extracts (40 µg) were prepared in SDS-PAGE sample buffer (25 mM Tris-HCl (pH 6.8), 2.5% β-mercaptoethanol, 1% SDS, 10% glycerol, 1 mM EDTA, 0.05 mg/ml bromophenol blue), heated to 95 °C for 5 min, and separated on 4–12% Tris-glycine-SDS gels (Fisher Scientific UK, Loughborough, UK). Proteins were then transferred onto an Immobilon FL PVDF membrane (Millipore, Watford, UK), blocked using Odyssey blocking buffer (Li-cor Biosciences, Cambridge, UK) and incubated with the appropriate primary antibody overnight at 4 °C. Membranes were washed with PBS containing 0.1% Tween 20, incubated with either Alexa Fluor 680 or IR Dye 800 secondary antibodies for 1 h at room temperature, and further washed with PBS containing 0.1% Tween 20. Proteins were visualized using the Odyssey Classic Infrared Imaging System (Li-cor Biosciences, Cambridge, UK).

### 4.4. Clonogenic Assays 

Cells were seeded in 35 mm dishes to ~70% confluence at the time of irradiation. Following irradiation, cells were trypsinized, counted and seeded into 6 well plates, and colonies were allowed to grow for 10–14 days, prior to fixing and staining with 6% glutaraldehyde and 0.5% crystal violet for 30 min. Plating efficiencies for the non-irradiated cells were as follows: OMM1 (~25%), OMM2.5 (~15%), Mel270 (~35%) and 92.1 (~40%). Dishes were washed, left to air dry overnight, and colonies were counted using the GelCount colony analyzer (Oxford Optronics, Oxford, UK). Relative colony formation (surviving fraction) was expressed as colonies per treatment level versus colonies which appeared in the untreated control. For inhibition experiments, media containing 10 µM ATM inhibitor (KU-55933; Selleck Chemicals, Munich, Germany) or DMSO as a vehicle only control were added onto the monolayer cells for 1 h prior to irradiation. The irradiated cells were then trypsinized, counted and seeded into 6 well plates containing 10 µM ATM inhibitor for a further 24 h. Following this, fresh media only was added until the colonies formed. Statistical analysis was performed using the CFAssay for R package [17].

### 4.5. Neutral Comet Assays

Neutral single cell gel electrophoresis (comet) assays for the measurement of DNA DSBs were performed as previously described [27,29,30]. Exponentially growing cells in 35 mm dishes were trypsinized, diluted to 1 × 10^5^ cell/ml, and 250 µl aliquots of the cell suspension were placed in the wells of a 24 well plate. For inhibitor experiments, media containing 10 µM ATM inhibitor (KU-55933) or DMSO as a vehicle only control was added to the cells 1 h prior to trypsinization. Cells in suspension were irradiated with 4 Gy X-rays or protons, and then embedded on a microscope slide in 1% low melting point agarose (Bio-Rad, Hemel Hempstead, UK) in PBS. Slides were incubated for up to 4 h in an humidified chamber to allow for DNA repair, and subsequently cell lysis was performed in buffer containing 2.5 M NaCl, 100 mM EDTA, 10 mM Tris-HCl pH 10.5, 1% N-lauroylsarcosine, 1% DMSO and 1% (*v*/*v*) Triton X-100 for at least 1 h at 4 °C. DNA was allowed to unwind in cold electrophoresis buffer (1 × TBE, pH 8.3) for 30 min in the dark, before separation of the DNA DSBs by electrophoresis at 25 V, ~20 mA for 25 min. Slides were washed three times with 1 × PBS before being allowed to air dry overnight. The agarose was subsequently rehydrated for 30 min in water (pH 8.0), stained for 30 min with SYBR Gold (Life Technologies, Paisley, UK) diluted 1:10,000 in water (pH 8.0), and air dried again overnight. Cells (50 per slide, in duplicate) were analyzed from the dried slides using the Komet 6.0 image analysis software (Andor Technology, Belfast, Northern Ireland), and three individual biological replicates were performed.

### 4.6. Immunohistochemistry

Sections (4 µm thick) from FFPE UM were deparaffinized and underwent heat induced epitope retrieval at 96 °C, pH 9.0 for 20 min on the Leica Bond RXm, followed by staining using the Leica Bond polymer refine red detection kit, according to the standard manufacturer’s protocols. A mouse monoclonal anti-ATM antibody (ab78; Abcam, Cambridge, UK) was used at 1:100 dilution. Sections were counter-stained with hematoxylin, dehydrated and mounted using a solvent based mountant. Negative controls used mouse IgG1 at the same concentration as the primary antibody. Staining was assessed and scored by three individuals (R.N.H., S.E.C., and J.K.). Statistical analysis was performed using a one sample *t*-test.

## 5. Conclusions 

We have demonstrated that UM cell lines display a broad range of radiosensitivity to photon (X-ray) radiotherapy and proton beam therapy, with an apparent correlation of increased radioresistance with elevated levels of ATM. Whilst no correlation between ATM protein expression in UM tissue samples and an adverse patient outcome was found, inhibition of ATM in vitro was shown to be effective in potentiating the effects of photons and protons in reducing clonogenic survival. Our data therefore suggest that targeting ATM, in combination with radiotherapy, can be a more effective strategy for the treatment of UM.

## Figures and Tables

**Figure 1 cancers-12-01388-f001:**
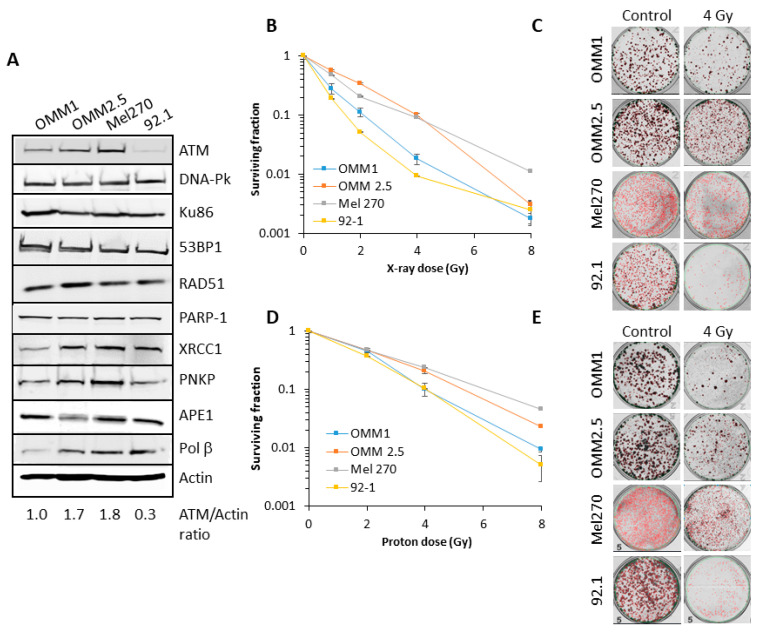
Comparative radiosensitivity of UM cells in response to photons and protons. (**A**) Whole cell extracts from UM cells were prepared and analyzed by immunoblotting with the indicated antibodies. Clonogenic survival of UM cells following treatment with increasing doses of (**B**–**C**) X-rays or (**D**)–(**E**) protons was analyzed from three independent experiments. (**B**) and (**D**) Shown is the surviving fraction ± S.E. (**C**) and (**E**) Representative images of colonies in non-irradiated and irradiated plates (the latter were seeded with double the number of cells, accordingly).

**Figure 2 cancers-12-01388-f002:**
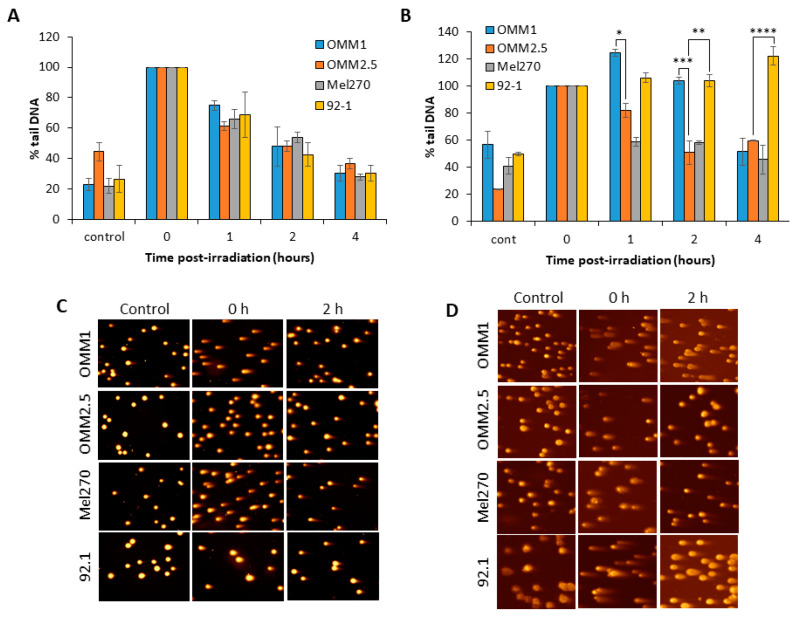
Repair of DSBs in UM cells in response to X-ray and proton irradiation. Cells were irradiated with (**A**) X-rays or (**B**) protons at 4 Gy, and DNA DSBs were measured at various time-points post-irradiation by neutral comet assays. Shown is the mean % tail DNA ± S.D. from at least three independent experiments, normalized to the levels seen immediately post-irradiation (time 0), which was set to 100%. Representative comet images in non-irradiated cells, and 0 or 2 h post-irradiation with either (**C**) X-rays or (**D**) protons. **p* < 0.05, ***p* < 0.01, ****p* < 0.005, *****p* < 0.001 by one sample *t*-test comparing OMM2.5 to OMM1 or 92.1.

**Figure 3 cancers-12-01388-f003:**
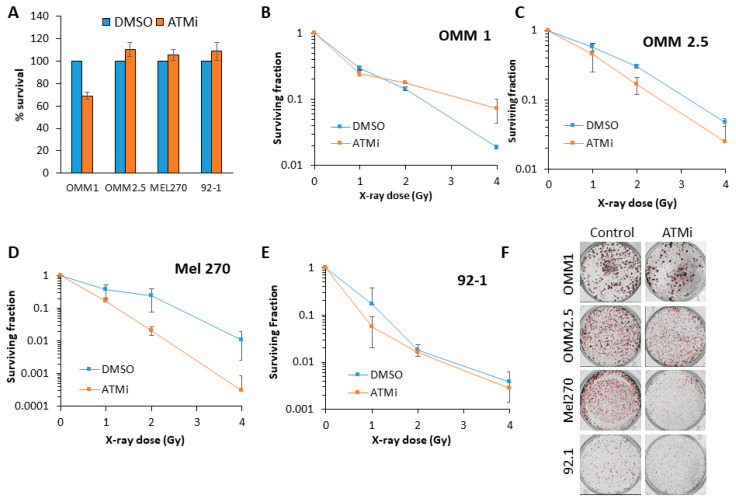
Comparative radiosensitivity of UM cells in the presence of ATM inhibition, following X-ray irradiation. The clonogenic survival of UM cells with (**A**) ATM inhibitor alone, or (**B**–**F**) following treatment with increasing doses of X-rays was analyzed from three independent experiments. (**B**–**E**) Shown is the surviving fraction ± S.E. (**F**) Representative images of colonies in non-irradiated and irradiated (2 Gy) plates (the latter were seeded with double the number of cells, accordingly).

**Figure 4 cancers-12-01388-f004:**
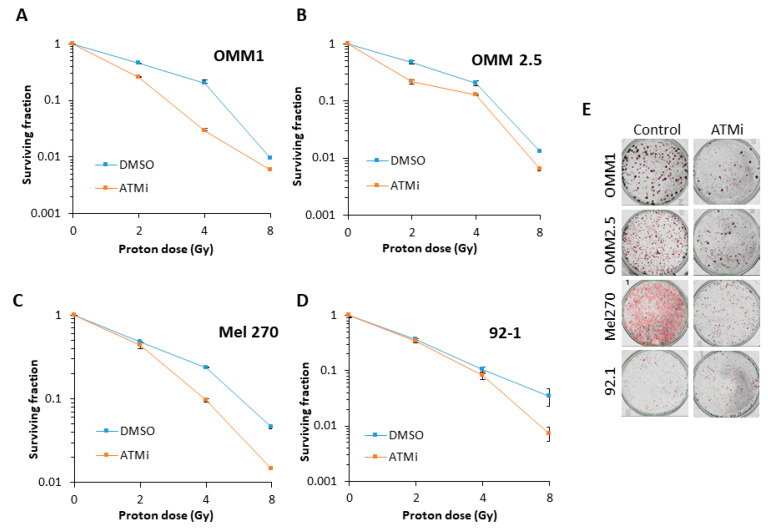
Comparative radiosensitivity of UM cells in the presence of ATM inhibition, following proton irradiation. The clonogenic survival of UM cells with ATM inhibition in the presence of increasing doses of protons was analyzed from three independent experiments. (**A**–**D**) Shown is the surviving fraction ± S.E. (**E**) Representative images of colonies in non-irradiated and irradiated (4 Gy) plates (the latter were seeded with double the number of cells, accordingly).

**Table 1 cancers-12-01388-t001:** Comparative survival of UM cells in response to photons and protons.

Cell Line	Comparative Cell Line (X-rays)	Statistical Analysis	Comparative Cell Line (Protons)	Statistical Analysis
92.1	OMM1	*p* = 0.07	OMM1	*p* = 0.98
92.1	OMM2.5	*p* < 0.0002	OMM2.5	*p* < 0.004
92.1	Mel270	*p* < 0.00001	Mel270	*p* < 0.003

Statistical analysis was performed using the CFAssay for R package [17], taking into account changes in survival across the complete dose response curves.

**Table 2 cancers-12-01388-t002:** Impact of ATM inhibition on the radiosensitivity of UM cells in response to photons and protons.

Cell Line	Statistical Analysis (Photons)	Statistical Analysis (Protons)
OMM1	*p* < 0.02*	*p* < 0.00001
OMM2.5	*p* < 0.05	*p* = 0.28
Mel270	*p* < 0.00001	*p* < 0.02
92.1	*p* = 0.81	*p* < 0.05

Statistical analysis was performed using the CFAssay for R package [17], taking into account changes in survival across the complete dose response curves. Note that * refers to increased radioresistance.

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
