# Peer review of "Inhibition of ATM Increases the Radiosensitivity of Uveal Melanoma Cells to Photons and Protons"

_cancers, 2020, doi:10.3390/cancers12061388_

Round 1

Reviewer 1 Report

Summary

In this paper, Hussain et al. perform photon and proton radiosensitivity assays on 4 uveal melanoma cell-lines and correlate these results with DNA repair protein expression. They discovered increased radiosensitivity in cell-lines with low ATM expression, and found that an ATM inhibitor can reverse radio-resistance. The authors should be commended for such a study in a rare cancer with translational applications.

Major points

1) TP53 mutations or deletions are strongly associated with radiation sensitivity. Please comment on TP53 mutation and copy-number status for the cell-lines. From Griewank et al. (“Genetic and molecular characterization of uveal melanoma cell lines”,2012), it does seem like these cell-lines likely do not harbor TP53 mutations. However, one should consider a TP53 blot for inclusion in Fig. 1A.

2) Small molecule kinase inhibitors are known to have off-targets. How are you certain that the inhibitor works as expected in your context? Please consider either using other means to perturb ATM (i.e. another inhibitor, siRNA, CRISPR, etc.) or blotting for downstream phosphorylation sites (i.e. phosopho-CHK2, phospho-TP53, etc.) following ATM inhibition.

Minor points

1) Line 87: “we firstly…” should read “we first”

2) For Table 1 and Table 5: How were the p-values calculated? From which dose level?

3) Section 2.3 is best left for the supplemental as the results do not reach significance.

Reviewer 2 Report

In this manuscript, the authors investigate the role of one of the major enzyme in the cellular response to DNA damage in radiosensitivity of Uveal Melanoma. One interest of this work is that  they analyse the cell survival and DNA repair activity after two types of irradiation using protons or photons. From a very reduced sample (4 cell lines) the authors identify the cellular concentration of ATM as a marker correlating with the radioresistance of the cells. Though such observations have already been published in other type of cancer (in particular Glioblastoma) there is no citation nor reference to the work already done on this subject. With the use of an ATM inhibitor the authors demonstrate that inhibition of ATM radiosensitize UM that express high level of ATM and do not sensitize cell lines with low level of ATM. The results are independent of the type of irradiation. The data seem enough solid to be published. However, an effort should be done in the discussion of results that were per se expected from the large bibliography on the subject.

Specific points:

  • In the abstract, “we also describe …necrotic reaction”. There is no real study on these topics. The Tables 2,3,4 are not discussed and seem to be part of another study. Please comment these data in the results or do not present
  • Line 96: the cell lines derived from primary and metastatic tumors have not been introduced
  • Figure 1: there is an important difference in the response to protons and photons of the different cell lines at low doses (2Gy) which decrease at 4Gy and disappear at 8Gy. This data should be discussed and taken in account in the comments of the Comet assay (Figure 2) which was performed within this range of dose. The protocol irradiating the cells in liquid before embedding does not allow to estimate the number of damage at time 0 since several minutes are required for the gel preparation with DNA repair still in process. A protocol with irradiation of already embedded cells would be more appropriate. Actually, the size of the comets at time 0h after photons seems already to be extremely reduced in some cell lines and therefore data normalized to this value could affect the interpretation.
  • 1 are the most sensitive to photons and protons but they show a DNA repair defect only after protons. An alkaline comet assay should have been performed to check if a difference in oxidative damage repair could explain this discrepency.
  • Table 2,3,4 are not discussed in the results

In table 4 the number of patients analyzed should be indicated for each caracteristics

  • Figure 4, 5. The dose and the scales are different in both figures. If the analyses is restricted to doses up to 4 Gy in both irradiation protocols the comments are wrong. The effect of ATMi is similar for photons and protons with no effect for 92-1 and strong inhibition for OMM1 and Mel270. OMM2.5 giving an intermediate result. Though the general conclusion remain the same, comments line 180-184 should be corrected. The fold-increase in radiosensitivity must be calculated at a same dose and precised in the text.
  • Discussion: p224 the lack of protein change for DNA-PK and KU might be due to the high cellular content of these proteins which impairs the accurate measurement of differences.
  • In a general way, there is no reference to the fact that ATM phosphorylation and nuclear relocation has been already proposed to predict radiosensitivity. In a general way, taking in account the huge literature on ATM phosphorylation a few sentences on this regulation of ATM should be added to this manuscript.

Round 2

Reviewer 1 Report

Authors have addressed all relevant concerns.